# OpenReview forum: "From Accuracy to Robustness: A Study of Rule- and Model-based Verifiers in Mathematical Reasoning"
_ICLR.cc/2026/Conference — ICLR 2026 Conference Withdrawn Submission_

### Official Review · Reviewer_KCtg · 2025-10-28

**Soundness:** 2
**Presentation:** 3
**Contribution:** 3
**Rating:** 4
**Confidence:** 4

**Summary:**

This paper investigates the reliability of verifiers: rule-based v.s. model-based in RLVR for mathematical reasoning. The authors evaluate existing rule-based verifiers across different training datasets, finding that their recall could drop compared with rigid equivalence checks. Thus the authors propose model-based and hybrid verifiers that combine rule-based precision with LLM-based flexibility. The authors also show that model-based verifiers yield +2–3 point gains in RL performance, but they introduce reward hacking challenging: policy models exploit verifier weaknesses to gain inflated rewards.

**Strengths:**

1. Verifier reliability is practically and conceptually important in RL.

2. The static and dynamic analyses span multiple open-source verifiers and datasets, revealing concrete recall-precision trade-offs.

3. The paper goes beyond accuracy metrics, exposing vulnerabilities of fine-tuned verifiers and proposing reward hacks.

**Weaknesses:**

1. In Figure 1, the differences among rule-based, verifier-based, and oracle reward curves are relatively minor. Table 2 further shows that the hybrid or model-based verifiers yield only about +2 points over the baseline. It is unclear whether such modest gains justify the additional computational and implementation overhead of integrating verifiers into the RL loop.

2. The curves labeled as verifier-hacked and non-hacked in Figure 1 are almost overlapping except at the very last step. This makes it difficult to attribute the observed performance drop to reward hacking rather than to stochastic variation in RL training. It is hard to see obvious performance effects of reward hacking to the RL training.

3. All experiments are conducted on Qwen-based models. The absence of results on other models limits the generality of the conclusions about verifier reliability and reward-hacking behavior.

4. Unclear value proposition of verifiers. The paper claims that introducing verifiers improves reward reliability, yet it simultaneously shows that verifiers increase the risk of reward hacking. Without a clear demonstration that the verifier cost is outweighed by significant performance improvements, it is hard to be convinced that verifiers “deserve” the added overheads. A more compelling storyline might instead argue that naive verifier design will be easily hacked, and then propose a new, more robust verifier design to tackle with this new challenging.

**Questions:**

1. Could the authors provide a cost–benefit analysis to quantify the extra cost of hybrid or model-based verifiers versus the performance gain reported in Table 2?

2. Do similar verifier behaviors hold for non-Qwen models?

3. Since verifiers can both improve reward recall and introduce hacking risk, could the authors propose a design principle that preserves reward reliability without exposing new vulnerabilities? This would make the paper more convincing and complete.

---

> ### Author Response · Authors · 2025-11-28
>
> Thank you for the review! We appreciate your comments, and we try to address your concerns in the response below.
>
> > W1: In Figure 1, the differences among rule-based, verifier-based, and oracle reward curves are relatively minor. Table 2 further shows that the hybrid or model-based verifiers yield only about +2 points over the baseline. It is unclear whether such modest gains justify the additional computational and implementation overhead of integrating verifiers into the RL loop.
>
> Thank you for raising this concern. We address it from two key perspectives:
> 1.  As shown in Figure 1, the hybrid verifier consistently outperforms the rule-based verifier across nearly all benchmarks. Although the overall average gain of 2.3 points is modest, we observe substantial improvements on specific tasks. For example, +5.1 points on AIME24 and +3.7 points on Minerva Math when trained on DeepScaleR, and +6 points on AMC23 when trained on Skywork-OR1. These gains are meaningful, especially considering that these datasets already achieve >88% recall with rule-based verifiers. The improvements stem from better utilization of complex  training samples -- cases that rule-based verifiers tend to misjudge or discard.
> 2. While the misjudged cases may account for only ~10% of responses in DeepScaleR and Skywork, we observe that this small fraction can already cause notable performance degradation. This impact becomes even more pronounced on more diverse datasets. For example, the HF verifier achieves only a 47% recall rate on WebInstruct-verified data. And following the setting in Appendix J, we further train Qwen-2.5-7B with rule-based and general verifiers on this dataset, comparing the performance of rule-based and general verifiers over 300 training iterations.  Evaluation on the GPQA-Diamond benchmark, presented below, demonstrates that the resulting performance gap is substantial.
>
> | Setting             |   GPQA-Diamond |
> |------------------|:----:|
> |  RL w/  HF verifier    | 41.4  |
> | RL w/ general-verifier    | 45.0  |
> | RL w/ R1-Verifier-1.5B (Hacking)   | 35.9  |
>
> In short, to maximize the effectiveness of RL with verifiable rewards, robust verifier design is critical. We believe the added computational cost is justified given consistent gains and long-term benefits, especially for complex domains, diverse data, and stronger policy models (Section 3).
>
> > W2: The curves labeled as verifier-hacked and non-hacked in Figure 1 are almost overlapping except at the very last step. This makes it difficult to attribute the observed performance drop to reward hacking rather than to stochastic variation in RL training. It is hard to see obvious performance effects of reward hacking to the RL training.
>
> We agree that at first glance, the verifier-hacked and non-hacked curves in Figure 1 may appear similar for much of the training process -- partly due to the scale of the y-axis, which compresses visible differences. However, Table 2 provides a clearer comparison:
> - The hacked setting achieves 55.6, which regresses toward the rule-based verifier performance (55.0).
> - The non-hacked setting reaches 57.3, a +1.7 point gain.
>
> This degradation is expected and aligns with the underlying dynamics of the RL training process. Specifically, when reward hacking occurs, it disrupts training by causing the GRPO algorithm to skip or downweight large portions of the data.
>
> Concretely, if the policy model begins generating hacked responses (e.g., trivial or deceptive completions) for certain queries, these queries end up receiving very low or even zero normalized rewards across all rollouts. As a result, they contribute little to no gradient signal and are effectively ignored during training.
>
> Although this may not manifest as a sharp drop in the training curve, it leads to early stagnation and prevents further improvements, an effect we interpret as a symptom of reward hacking. In this sense, the absence of expected growth serves as a key signal of hacking, rather than sudden deterioration.

---

> ### Author Response · Authors · 2025-11-28
>
> > W3: All experiments are conducted on Qwen-based models. The absence of results on other models limits the generality of the conclusions about verifier reliability and reward-hacking behavior.
>
> Our study primarily focuses on verifying the robustness of different verifiers across datasets (DeepScaleR, Skywork-OR1, WebInstruct-Verified) and domains (mathematical and general scientific reasoning), which we believe are the more critical dimensions of generalization in the context of verifiable rewards.
>
> While our RL experiments focus on a single model (Qwen-2.5-7B) due to computational constraints, we conducted extensive static evaluations in Section 3 using four models, ranging from 7B to 32B, and spanning both reasoning and non-reasoning architectures.
>
> These results highlight a key trend: as model capabilities increase, the limitations of rule-based verifiers become significantly more pronounced. This provides strong evidence that performance gaps between rule-based and hybrid verifiers (in the absence of reward hacking) are likely to widen with more powerful models. At the same time, stronger policy models are also more likely to discover and exploit vulnerabilities in model-based verifiers, leading to more severe reward hacking.
>
> > W4: Unclear value proposition of verifiers. The paper claims that introducing verifiers improves reward reliability, yet it simultaneously shows that verifiers increase the risk of reward hacking. Without a clear demonstration that the verifier cost is outweighed by significant performance improvements, it is hard to be convinced that verifiers “deserve” the added overheads. A more compelling storyline might instead argue that naive verifier design will be easily hacked, and then propose a new, more robust verifier design to tackle with this new challenge.
>
> Thank you for the thoughtful question. This point is closely related to our response to W1.
> As demonstrated in Figure 1, the hybrid verifier consistently outperforms the rule-based verifier, and importantly, this performance gap does not converge with additional compute. This suggests that scaling alone is insufficient. And introducing a stronger verifier is necessary to achieve higher performance.
>
> Furthermore, in more diverse and less structured datasets like WebInstruct, the limitations of rule-based verifiers become even more pronounced. This highlights that verifier robustness is not optional in such settings -- it is essential.
>
> That said, we acknowledge your point: while verifiers improve reward reliability, they also introduce new vulnerabilities to reward hacking. Our paper is intentionally focused on exposing this trade-off, demonstrating both the value and the limitations of current verifier designs. We view this as a necessary first step: before proposing solutions, it is critical to understand and quantify the problem.
>
> You’re right that a natural next step is to design more robust verifiers that can resist hacking while preserving accuracy. We hope that our findings will motivate and guide future work in that direction.
>
>
>
>
> > Q1: Could the authors provide a cost–benefit analysis to quantify the extra cost of hybrid or model-based verifiers versus the performance gain reported in Table 2?
>
> In our responses to W1 and W4, we demonstrate that the performance gap between the hybrid verifier and the rule-based verifier does not diminish with additional compute cost; in fact, it widens on the WebInstruct dataset. This indicates that verifier robustness is not optional in such settings -- rather, it is essential.
>
> Please refer to our detailed discussion in the responses to W1 and W4 for further justification.
>
>
> > Q2: Do similar verifier behaviors hold for non-Qwen models?
>
> Please refer to the response to W3 above： While our RL experiments focus on a single model (Qwen-2.5-7B) due to computational constraints, we conducted extensive static evaluations in Section 3 using four models, ranging from 7B to 32B, and spanning both reasoning and non-reasoning architectures.
>
> > Q3: Since verifiers can both improve reward recall and introduce hacking risk, could the authors propose a design principle that preserves reward reliability without exposing new vulnerabilities?
> This would make the paper more convincing and complete.
>
> Thank you for the question. Based on our findings, we propose the following actionable guidelines:
> 1. Use rule-based verifiers when they offer high precision and recall--they are typically sufficient and reliable for RL.
> 2. In domains lacking effective rule-based verifiers, consider hybrid or model-based verifiers depending on the domain constraints.
> 3. Crucially, when using model-based verifiers, assess not only static accuracy but also robustness during RL.

---

### Official Review · Reviewer_JfKs · 2025-10-31

**Soundness:** 2
**Presentation:** 1
**Contribution:** 2
**Rating:** 2
**Confidence:** 4

**Summary:**

This paper analyzes the verifier component of RLVR in mathematical reasoning tasks. The authors point out that commonly used rule-based verifiers buffer from high precision but low recall, as they perform well only on responses following specific patterns. To mitigate this limitation, LLMs are utilized as a model-based verifier based on its reasoning skills.
Furthermore, the authors proposed a hybrid verifier and also analyzed the reward hacking  problem in using trained model-based verifiers.

**Strengths:**

* The paper attempts to clarify and analyze key issues often overlooked in RLHF and RLVR research, particularly the limitations of rule-based versus model-based verifiers. This focus addresses an important and timely problem in the field.

* The discussion on potential reward hacking and robustness issues arising from the use of model-based verifiers is interesting, providing new perspectives on challenges that are often underexplored in current research.

**Weaknesses:**

**\[W1\] Insufficient Analysis**
The paper's main motivation is that the impact of verifier types on RLVR is poorly understood, yet it lacks in-depth analysis on this topic. There is no error case analysis explaining why rule- and model-based verifiers fail, nor any examination of how these failures influence policy behavior. Without these critical components, the paper lacks the insights necessary to address its core motivation.

**\[W2\] Low Readability**
The overall organization of the paper lacks clarity, which makes it difficult to follow the intended narrative. In the abstract and introduction, the authors state that the paper focuses on a comparative analysis between rule-based verifiers and model-based verifiers. However, there is no mention of hybrid verifiers in these sections. Starting from Section 3.3, the paper significantly discusses hybrid verifiers, but the motivation for introducing this concept and its role within the paper’s main objective are not clearly explained. As a result, the overall flow of the paper becomes confusing, and readers may struggle to understand how each section contributes to the central argument.

In addition, the placement of figures and tables often lacks alignment with the corresponding text. Some figures (e.g., Figure 1\) combine content drawn from multiple sections, which makes cross-checking difficult.

**\[W3\] Limited Diversity of Verifiers and Policy Models**
The hybrid setting is tested with only three model-based verifiers, leaving unclear whether the reported vulnerabilities generalize across architectures or scales. Moreover, RL training uses a single policy (Qwen 2.5 7B), which restricts analysis of verifier–policy interactions. Broader experiments with different verifier families and policy sizes would provide stronger evidence for the claimed trends.

**Questions:**

**\[Q1\]** Lines 398-399: What explains the instability observed in trained verifier models compared to untrained verifiers and rule-based verifiers? It seems counterintuitive that untrained model-based verifiers don't exhibit the same instability as their trained counterparts.

**\[Q2\]** Given that GPT-4o was utilized as the oracle reward, what performance metrics would this model achieve if implemented directly as a model-based verifier? If it demonstrates high precision and recall, wouldn't this suggest that employing LLMs with superior reasoning capabilities is a more direct solution?

**\[Q3\]** Are models with strong reasoning capabilities like GPT-4o also susceptible to reward hacking? This question has implications for the fundamental approach proposed in the paper.

---

> ### Author Response · Authors · 2025-11-28
>
> Thank you for the review! We appreciate your comments, and we try to address your concerns in the response below.
>
> > W1: The paper's main motivation is that the impact of verifier types on RLVR is poorly understood, yet it lacks in-depth analysis on this topic. There is no error case analysis explaining why rule- and model-based verifiers fail, nor any examination of how these failures influence policy behavior. Without these critical components, the paper lacks the insights necessary to address its core motivation.
>
> For the rule-based verifier, we present two representative false negative examples in Figure 5 (Appendix D) , and explicitly reference them in Line 190 of Section 3.2. For the reward hacking phenomenon, we include two concrete cases in Figure 11 and Figure 12 (Appendix L), and provide a dedicated discussion in Lines 406. All line numbers are updated to match the latest revision, but the corresponding content was already present in the original version. Could you please check again?
>
> > W2: The overall organization of the paper lacks clarity, which makes it difficult to follow the intended narrative. In the abstract and introduction, the authors state that the paper focuses on a comparative analysis between rule-based verifiers and model-based verifiers. However, there is no mention of hybrid verifiers in these sections. Starting from Section 3.3, the paper significantly discusses hybrid verifiers, but the motivation for introducing this concept and its role within the paper’s main objective are not clearly explained. As a result, the overall flow of the paper becomes confusing, and readers may struggle to understand how each section contributes to the central argument. In addition, the placement of figures and tables often lacks alignment with the corresponding text. Some figures (e.g., Figure 1) combine content drawn from multiple sections, which makes cross-checking difficult.
>
> To clarify, in this paper we evaluate model-based verifiers within a hybrid framework -- that is, rule-based verifiers are applied first, and only when they fail to verify a response do we invoke model-based verifiers. The motivation is to filter out trivial cases that rule-based verifiers can reliably handle, allowing model-based verifiers to focus on non-trivial or ambiguous cases. This design choice is explicitly discussed in Lines 246–251 and elaborated in Section 4.1.
> Thank you also for your suggestion regarding Figure 1. We agree that placing the figure early (on page 2) while referring to it much later (on page 6) disrupted the narrative flow. In the revised version during rebuttal, we have relocated Figure 1 to Section 4 to better align with the surrounding discussion and improve readability.
>
> > W3: The hybrid setting is tested with only three model-based verifiers, leaving unclear whether the reported vulnerabilities generalize across architectures or scales. Moreover, RL training uses a single policy (Qwen 2.5 7B), which restricts analysis of verifier–policy interactions. Broader experiments with different verifier families and policy sizes would provide stronger evidence for the claimed trends.
>
> In our RL experiments, we include three model-based verifiers selected to be representative within their respective categories (under the 1.5B scale):  the best General LLM (DS-R1-Distill-Qwen-1.5B), the best trained verifier (general-verifier) and our own trained verifier. For static evaluation (Section 3), we include a much broader range of verifiers across different model sizes and sources. However, due to computational constraints, we are unable to conduct full RL experiments for all of them. We believe the selected subset is sufficiently representative to support our key conclusions.
>
> Our study primarily focuses on verifying the robustness of different verifiers across datasets (DeepScaleR, Skywork-OR1, WebInstruct-Verified) and domains (mathematical and general scientific reasoning), which we believe are the more critical dimensions of generalization in the context of verifiable rewards.
>
> While our RL experiments focus on a single model (Qwen-2.5-7B) due to computational constraints, we conducted extensive static evaluations in Section 3 using four models, ranging from 7B to 32B, and spanning both reasoning and non-reasoning architectures.
>
> These results highlight a key trend: as model capabilities increase, the limitations of rule-based verifiers become significantly more pronounced. This provides strong evidence that performance gaps between rule-based and hybrid verifiers (in the absence of reward hacking) are likely to widen with more powerful models. At the same time, stronger policy models are also more likely to discover and exploit vulnerabilities in model-based verifiers, leading to more severe reward hacking.

---

> ### Author Response · Authors · 2025-11-28
> **Responses to the review (2/2)**
>
> > Q1:  Lines 398-399: What explains the instability observed in trained verifier models compared to untrained verifiers and rule-based verifiers? It seems counterintuitive that untrained model-based verifiers don't exhibit the same instability as their trained counterparts.
>
> Good question. We believe that trained verifiers are more vulnerable to attack compared to general model-based verifiers, because task-specific fine-tuning can intuitively reduce a model’s generalization and robustness, potentially leading to overfitting on the training data. In contrast, general model-based verifiers are not trained for a specific task but are instead exposed to diverse tasks during large-scale post-training, which generally makes them more robust.
>
> > Q2: Given that GPT-4o was utilized as the oracle reward, what performance metrics would this model achieve if implemented directly as a model-based verifier? If it demonstrates high precision and recall, wouldn't this suggest that employing LLMs with superior reasoning capabilities is a more direct solution?
>
> Good question, we agree that using a strong model like GPT-4o as a verifier would likely yield higher precision and recall, and the performance gap between the policy model and verifier is indeed an important dimension worth further study. However, in practice, using a significantly stronger model as the verifier is often infeasible. Because relying on a verifier stronger than the policy would preclude training models that aim to surpass GPT-4o, which contradicts the goals of frontier development. In short, while GPT-4o provides a useful oracle for evaluation, our study aims to address the practical challenges of training with verifiers that are comparable to or weaker than the policy model--an inevitable scenario when pushing model capabilities forward.
>
> > Q3: Are models with strong reasoning capabilities like GPT-4o also susceptible to reward hacking? This question has implications for the fundamental approach proposed in the paper.
>
> Yes, absolutely. Even models with advanced reasoning capabilities—such as GPT-4o -- can be vulnerable to reward hacking when supervising stronger models. For example, in the work by OpenAI [1], GPT-4o was used as a monitoring agent in agentic coding environments, with the goal of detecting misbehavior. The authors found that even strong models like GPT-4o can be fragile in such settings. This aligns with our broader point: stronger verifiers do not guarantee immunity to reward hacking, particularly in dynamic RL environments where policy models can adapt in unexpected ways.
>
> [1] Baker, Bowen, et al. "Monitoring reasoning models for misbehavior and the risks of promoting obfuscation." arXiv preprint arXiv:2503.11926 (2025).

---

### Official Review · Reviewer_7SQK · 2025-10-31

**Soundness:** 3
**Presentation:** 2
**Contribution:** 2
**Rating:** 4
**Confidence:** 4

**Summary:**

The paper investigates verifier accuracy and its impact on model performance in reinforcement learning with verifiable rewards (RLVR) for mathematical reasoning tasks. Through systematic experiments, the authors show that commonly used rule-based verifiers, while highly precise, suffer from false negatives that lead to suboptimal training outcomes. They further evaluate model-based verifiers and find that, although recall improves, these verifiers are vulnerable to reward hacking during RL training. To mitigate this issue, the paper proposes a hybrid verifier that combines rule-based and model-based verifications, achieving greater stability and performance. Overall, the study underscores the critical role of verifier design in ensuring reliable reward signals and provides some practical guidance for building more robust verifier systems.

**Strengths:**

- **Originality.** The paper provides a systematic and timely investigation of verifier design in RL with verifiable rewards (RLVR), providing one of the first comprehensive analyses of how verifier accuracy impacts training stability and model performance and exposes limitations in current verification systems.
- **Comprehensive experimentation.** The study conducts extensive experiments comparing rule-based and model-based verifiers, builds dedicated diagnostic datasets, and performs multiple RL training runs with the Qwen2.5-7B model under different verifier configurations.
- **Quality.** The experimental design is sound and rigorous, with consistent metrics and well-documented procedures. The reported improvements are clear and no apparent flaws are evident in the setup and evaluation process.

**Weaknesses:**

- **[Significance]** While the paper presents systematic experiments and insightful analyses, many of its findings confirm known issues rather than reveal fundamentally new phenomena. Specifically, (1) the false-negative problem of rule-based verifiers has been discussed in prior work on mathematical expression evaluation (e.g., [1], [2]); and (2) the vulnerability of LLM-based verifiers to reward hacking aligns with broader findings on LLM-as-a-judge robustness and the reward hacking in RLHF (e.g., [3], [4]). Although this paper is among the first to document such hacking behaviors in verifier-based RL, the results are largely predictable. The proposed hybrid strategy, which combines rule-based and model-based verifiers, is practical but conceptually straightforward, as both components are adapted from existing methods.
- **[Experiment]** The RL evaluation primarily uses a single policy model (Qwen2.5-7B). Without additional policy models, it is difficult to assess whether the observed verifier effects generalize across architectures or model scales.
- **[Presentation]** The paper’s presentation is somewhat disorganized, making it easy for readers to lose track of the experimental narrative. Numerous experimental setups are scattered throughout the text, and the analyses are often separated from their corresponding results. For example, Figure 1 is introduced early (page 2) but its related experiment is not discussed until page 6; likewise, the analysis of the hybrid verifier in Section 4.1 refers to results that only appear in the appendix. Additionally, Sections 5.2 and 6 contain overlapping analyses on the reward hacking patterns. Overall, this fragmented structure reduces readability and weakens the logical flow of the paper.

[1] Non-Autoregressive Math Word Problem Solver with Unified Tree Structure. EMNLP 2023.

[2] TinyV: Reducing False Negatives in Verification Improves RL for LLM Reasoning. Arxiv 2025.

[3] Is LLM-as-a-Judge Robust? Investigating Universal Adversarial Attacks on Zero-shot LLM Assessment. EMNLP 2024.

[4] ODIN: Disentangled Reward Mitigates Hacking in RLHF. ICML 2024.

**Questions:**

- How do the authors interpret the noise in rule-based verifiers? Do such negative samples represent a fundamental limitation for developing better reasoning models, or could they instead be viewed as tolerable label noise—given that neural networks are often robust to noisy supervision [1, 2]? Moreover, since hybrid verification introduces additional training cost, why not explore simpler scaling strategies, such as increasing data volume or training steps, to mitigate the noise effect?
- A potentially insightful extension would be to quantify verifier noise tolerance in RLVR, i.e., how much noise the training can tolerate before noticeable performance degradation or collapse occurs. Do the authors have any empirical observations or insights on this aspect?

[1] Deep learning is robust to massive label noise. Arxiv 2017.

[2] Spurious Rewards: Rethinking Training Signals in RLVR. Arxiv 2025.

---

> ### Author Response · Authors · 2025-11-28
> **Responses to the review (1/2)**
>
> Thank you for the review! We appreciate your comments and we try to address your concerns in the response below.
>
> > W1: [Significance] While the paper presents systematic experiments and insightful analyses, many of its findings confirm known issues rather than reveal fundamentally new phenomena. ... Although this paper is among the first to document such hacking behaviors in verifier-based RL, the results are largely predictable. The proposed hybrid strategy, which combines rule-based and model-based verifiers, is practical but conceptually straightforward, as both components are adapted from existing methods.
>
> Thank you for raising this point. We agree that both the false negative (FN) issue of rule-based verifiers and the vulnerability of model-based verifiers to reward hacking have been discussed in prior literature. However, we would like to respectfully clarify our contributions in this context:
> 1. On rule-based verifier limitations: While the FN problem is conceptually intuitive and has been mentioned (e.g., [1], [2]), these works are not focused on verifier evaluation. For instance, [1] primarily explores how diverse mathematical expressions hinder learning, rather than analyzing verifiers or their interaction with RL systems. Meanwhile, [2] is concurrent work with our submission and evaluates only a narrow set of verifiers (e.g., Prime Verifier). In contrast, our study conducts a comprehensive empirical analysis of several widely adopted and high-performing verifiers (e.g., Qwen-Math, HF Verifier), which are more representative of latest RL applications.
> 2. On model-based verifier vulnerabilities: While reward hacking has been long recognized in RL, not just RLHF, its concrete manifestation and mechanisms in the RLVR setting remain underexplored. Our work is, to our knowledge, the first to systematically quantify reward hacking under RLVR using oracle-based analysis (via GPT-4o) and trace its dynamics during training. Furthermore, we uncover a non-trivial insight that fine-tuned model-based verifiers are more vulnerable to reward hacking than general LLM judges -- a finding that directly challenges the common practice of using task-specific verifier fine-tuning for RL.
> 3. On the hybrid verifier design: We agree with the reviewer that the hybrid verifier design is straightforward and do not claim it as a primary contribution. Instead, we view it as a practical experimental settings for our analysis.
>
> > W2: [Experiment] The RL evaluation primarily uses a single policy model (Qwen2.5-7B). Without additional policy models, it is difficult to assess whether the observed verifier effects generalize across architectures or model scales.
>
> Our study primarily focuses on verifying the robustness of different verifiers across datasets (DeepScaleR, Skywork-OR1, WebInstruct-Verified) and domains (mathematical and general scientific reasoning), which we believe are the more critical dimensions of generalization in the context of verifiable rewards.
> While our RL experiments focus on a single model (Qwen-2.5-7B) due to computational constraints, we conducted extensive static evaluations in Section 3 using four models, ranging from 7B to 32B, and spanning both reasoning and non-reasoning architectures.
> These results highlight a key trend: as model capabilities increase, the limitations of rule-based verifiers become significantly more pronounced. This provides strong evidence that performance gaps between rule-based and hybrid verifiers (in the absence of reward hacking) are likely to widen with more powerful models. At the same time, stronger policy models are also more likely to discover and exploit vulnerabilities in model-based verifiers, leading to more severe reward hacking.

---

> ### Author Response · Authors · 2025-11-28
>
> > W3: [Presentation] The paper’s presentation is somewhat disorganized, making it easy for readers to lose track of the experimental narrative. Numerous experimental setups are scattered throughout the text, and the analyses are often separated from their corresponding results. For example, Figure 1 is introduced early (page 2) but its related experiment is not discussed until page 6; likewise, the analysis of the hybrid verifier in Section 4.1 refers to results that only appear in the appendix. Additionally, Sections 5.2 and 6 contain overlapping analyses on the reward hacking patterns. Overall, this fragmented structure reduces readability and weakens the logical flow of the paper.
>
> Thank you for this helpful suggestion. We have made the revisions to improve the presentation. First, we agree that introducing Figure 1 early (on page 2) while discussing it later (on page 6) disrupted the narrative flow. In the revision, we have moved Figure 1 to Section 4. Due to space limitations, we have kept the static evaluation results of the hybrid verifier in Appendix F, as they involve multiple detailed tables.
> Regarding the discussion of reward hacking patterns in Sections 5.2 and 6, we’d like to clarify that they serve distinct purposes:
> - In Section 5.2, we focus on a case study of reward hacking during RL training, illustrating how it occurs.
> - In Section 6, we  aims at evaluating the connection between artificially constructed attack patterns and the real behaviors observed during RL.
> We have revised Section 6 accordingly to make this distinction clearer. All updates have been applied to the revised manuscript and are marked in red.
>
> > Q1: How do the authors interpret the noise in rule-based verifiers? Do such negative samples represent a fundamental limitation for developing better reasoning models, or could they instead be viewed as tolerable label noise—given that neural networks are often robust to noisy supervision [1, 2]? Moreover, since hybrid verification introduces additional training cost, why not explore simpler scaling strategies, such as increasing data volume or training steps, to mitigate the noise effect?
>
> We believe that the false negatives (FNs) introduced by rule-based verifiers have a substantial impact on training frontier models, as they directly reduce data utilization efficiency. As shown in Section 3, this issue becomes increasingly severe as both model capability and data complexity increase, resulting in tangible effects on the RL training process, as illustrated in Figure 1.
> To further verify this hypothesis, we conduct additional experiments following the setting in Appendix J, where we train Qwen-2.5-7B on the WebInstruct-Verified dataset using both a HF verifier and a general model verifier, comparing their performance over 300 RL iterations. The results on the GPQA-Diamond benchmark are as follows:
>
> | Setting             |   GPQA-Diamond |
> |------------------|:----:|
> |  RL w/  HF verifier    | 41.4  |
> | RL w/ general-verifier    | 45.0  |
> | RL w/ R1-Verifier-1.5B (Hacking)   | 35.9  |
>
> As shown in Table 7 of Appendix J, the HF verifier achieves only a 47% recall on WebInstruct, significantly lower than in the math domain due to the dataset’s internal diversity. This leads to a performance gap of 3.6 points in RL outcomes -- demonstrating that the impact of FNs is not mitigated by increased data volume, and may worsen with greater data diversity. Furthermore, we note that increasing the number of training steps does not close this performance gap. As shown in Figure 1, the gap between rule-based and model-based (non-hacked) verifiers remains consistently large throughout training, suggesting that this is a systemic bottleneck rather than an artifact of early-stage instability.
> Lastly, regarding the paper [2] cited in the review: we note that its baseline is considerably weaker  than many other works (49.4 v.s 63.6 on MATH) such as SimpleRL[1] which raises concerns about its conclusions.
>
> [1] Zeng, Weihao, et al. "Simplerl-zoo: Investigating and taming zero reinforcement learning for open base models in the wild." arXiv preprint arXiv:2503.18892 (2025).
>
> > Q2: A potentially insightful extension would be to quantify verifier noise tolerance in RLVR, i.e., how much noise the training can tolerate before noticeable performance degradation or collapse occurs. Do the authors have any empirical observations or insights on this aspect?
>
> Thank you for this insightful question. We agree that quantifying the noise tolerance of verifiers in RLVR would be highly valuable. Our empirical results (e.g., Figure 1 and Appendix J) already suggest noticeable performance degradation with even modest levels of verifier noise (e.g., 10–15% false negatives), particularly in complex or diverse domains. We hope our findings provide a foundation for future work to systematically study noise tolerance under varying verifier properties and task settings.

---

### Official Review · Reviewer_oRgD · 2025-11-01

**Soundness:** 3
**Presentation:** 3
**Contribution:** 2
**Rating:** 2
**Confidence:** 4

**Summary:**

This paper analyzes the reliability of rule-based vs. model-based verifiers in reinforcement learning with verifiable reward (RLVR), focusing on mathematical reasoning tasks. Experimental results demonstrate that Rule-based verifiers achieve near-perfect precision but poor recall, increasingly misclassifying correct answers from stronger models. Model-based verifiers can offer higher recall and better flexibility across datasets, but are prone to reward hacking during RL training.

**Strengths:**

1. The paper is clearly written and easy to follow, with a logical structure and clear presentation of results.
2. It addresses an important and timely question about the reliability of verifiers in RL-based fine-tuning.
3. The work provides a detailed and systematic statistical analysis comparing rule-based and model-based verifiers across multiple benchmarks.

**Weaknesses:**

1. The paper is mostly empirical and lacks a formal analysis of why RL dynamics amplify verifier brittleness.
2. The study focuses almost exclusively on mathematical reasoning; generalization to other domains is less mentioned.
3. Reported gains in RL experiments are small and may not exceed noise given limited sampling. Statistical uncertainty isn’t reported.
4. The paper lacks a clear concluding message or actionable suggestion. While it identifies the limitations of both rule-based and model-based verifiers, it does not provide concrete guidance or a principled framework for designing more robust evaluation systems.
5. Section 6 feels unconvincing to me. The probing study on “hacking patterns” appears artificial and disconnected from realistic training dynamics. In practical RL settings, it is unclear how likely policy models are to autonomously discover and exploit such handcrafted adversarial patterns.

**Questions:**

1. Have the authors considered evaluating the findings on more general or non-mathematical reasoning tasks to test cross-domain robustness?
2. I am quite intersted in what mechanisms cause fine-tuned verifiers to become more vulnerable.
3. Could adversarially trained verifiers (such as through contrastive fine-tuning on generated hacking examples) help improve robustness without sacrificing recall?

---

> ### Author Response · Authors · 2025-11-28
> **Responses to the review (1/2)**
>
> Thank you for the review! We appreciate your comments, and we try to address your concerns in the response below.
>
> > W1: The paper is mostly empirical and lacks a formal analysis of why RL dynamics amplify verifier brittleness.
>
> In this work, we focus primarily on empirical identification and analysis of reward hacking phenomena. While we provide some observations and hypotheses, we acknowledge that a deeper theoretical understanding of why these behaviors emerge remains an open question and is left for future work.
>
> > W2: The study focuses almost exclusively on mathematical reasoning; generalization to other domains is less mentioned.
>
> We agree that cross-domain generalization is important. In addition to mathematical reasoning, **we would like to remind the reviewer that our original submission contained  further evaluation and RL training on the WebInstruct-Verified dataset, which spans a broad general science domain (Physics, Chemistry, Biology, Business, Finance).** These findings were discussed in Appendix J and explicitly referenced around Line 203 and Line 411. The Static Evaluation shows that rule-based verifiers exhibit even lower recall compared to the math domain, further reflecting their limited flexibility in handling diverse and less structured answer formats. And for the RL experiments, the training reward (from the verifier) and oracle reward (from GPT-4o) reveal a clear reward hacking pattern.  Following this point, downstream performance on GPQA-Diamond stagnates and even declines, further confirming the issue. These results and findings are consistent with the conclusions in the math domain.
>
> >W3: Reported gains in RL experiments are small and may not exceed noise, given limited sampling. Statistical uncertainty isn’t reported.
>
> We appreciate this comment  and would like to respond from two perspectives:
> 1.  As shown in Figure 1, the hybrid verifier consistently outperforms the rule-based verifier across nearly all benchmarks. Although the overall average gain of 2.3 points is modest, we observe substantial improvements on specific tasks. For example, +5.1 points on AIME24 and +3.7 points on Minerva Math when trained on DeepScaleR, and +6 points on AMC23 when trained on Skywork-OR1.  This consistency suggests the improvements are not merely noise.  Due to computational constraints, multiple full RL runs are infeasible, but we mitigate variance by evaluating AIME24 and AMC23 with Avg@32 rollouts.
> 2. Note that both DeepScaleR and Skywork reflect relatively lower, yet absolutely decent recall rates of >88% for rule-based verifiers. Consequently, the misjudgment only affects around 10% of the positive responses during RL training.  Despite this seemingly small fraction, we find it significant that such a limited portion can lead to the gaps observed in DeepScaleR and SkyWork. We believe that this performance gap is likely to widen for many other datasets. For example, the HF verifier achieves only a 47% recall rate on WebInstruct-verified data. And following the setting in Appendix J, we further train Qwen-2.5-7B with rule-based and general verifiers on this dataset, comparing the performance of rule-based and general verifiers over 300 training iterations.  Evaluation on the GPQA-Diamond benchmark, presented below, demonstrates that the resulting performance gap is substantial.
>
> | Setting |   GPQA-Diamond |
> |----|:----:|
> |  RL w/  HF verifier    | 41.4  |
> | RL w/ general-verifier    | 45.0  |
> | RL w/ R1-Verifier-1.5B (Hacking)   | 35.9  |
>
> > W4: The paper lacks a clear concluding message or actionable suggestion. ...
>
> Thank you for the question. Based on our findings, we propose the following actionable guidelines:
> 1. Use rule-based verifiers when they offer high precision and recall -- they are typically sufficient and reliable for RL.
> 2. In domains lacking effective rule-based verifiers, consider hybrid or model-based verifiers depending on the domain constraints.
> 3. Crucially, when using model-based verifiers, assess not only static accuracy but also robustness during RL.
>
> > W5: Section 6 feels unconvincing to me. ...it is unclear how likely policy models are to autonomously discover and exploit such handcrafted adversarial patterns.
>
> We appreciate this concern. The purpose of Section 6 is twofold:
> 1. First, it extends concrete hacking behaviors observed during RL (Section 5), such as “single symbol” and “gibberish” outputs.
> 2. Second, it systematically probes a broader range of adversarial patterns to reveal structural vulnerabilities in different types of verifiers.
> Our current policy model is a relatively weak 7B model. Stronger RL-trained models may discover additional or more sophisticated patterns. Therefore, Section 6 should be viewed as providing constructive diagnostic tools, rather than claiming that all patterns will emerge autonomously during training.

---

> ### Author Response · Authors · 2025-11-28
> **Responses to the review (2/2)**
>
> > Q1: Have the authors considered evaluating the findings on more general or non-mathematical reasoning tasks to test cross-domain robustness?
>
> Please refer to the response to W2 above.
>
> > Q2: I am quite interested in what mechanisms cause fine-tuned verifiers to become more vulnerable.
>
> This is a great question. We believe that trained verifiers are more vulnerable to attack compared to general model-based verifiers, because task-specific fine-tuning can intuitively reduce a model’s generalization and robustness, potentially leading to overfitting on the training data. In contrast, general model-based verifiers are not trained for a specific task but are instead exposed to diverse tasks during large-scale post-training, which generally makes them more robust.

---

### Note · Authors · 2026-01-06

I have read and agree with the venue's withdrawal policy on behalf of myself and my co-authors.